# Isavuconazole Pharmacokinetics and Pharmacodynamics in Children

**DOI:** 10.3390/pharmaceutics15010075

**Published:** 2022-12-26

**Authors:** Hirsh Elhence, Kanokporn Mongkolrattanothai, Sindhu Mohandas, Michael N. Neely

**Affiliations:** 1Keck School of Medicine, University of Southern California, Los Angeles, CA 90027, USA; 2Division of Infectious Diseases, Children’s Hospital Los Angeles, Los Angeles, CA 90027, USA; 3Laboratory of Applied Pharmacokinetics and Bioinformatics, The Saban Research Institute, Children’s Hospital Los Angeles, Los Angeles, CA 90027, USA

**Keywords:** isavuconazole, pediatric, pharmacokinetics, pharmacodynamics, Bayesian, therapeutic drug monitoring, fungal infection, outcomes

## Abstract

Isavuconazole is a broad-spectrum azole anti-fungal not yet approved in children. We conducted a retrospective, single-center review of isavuconazole use and routine therapeutic drug monitoring in pediatric patients, extracting demographic, dosing, concentration, mortality and hepatoxicity data. We constructed a nonparametric population model using Pmetrics. Of 26 patients, 19 (73%) were male. The mean (SD) age and weight were 12.7 (5.5) years and 50.9 (26.8) kg. Eighty percent received between 9.7 and 10.6 mg/kg per dose. Ten (38%) subjects had proven fungal disease and eight (31%) had probable disease, mostly with *Candida* and *Aspergillus* spp. The predicted steady-state isavuconazole concentrations in our patients were similar to previous reports in children and adults, and simulations with the proposed dosing of 10 mg/kg/dose every 8 h for 2 days followed by once daily maintenance matched effective adult exposures. Attributable mortality (5 of 11 deaths) was associated with steady-state daily AUC < 60 mg∗h/L and higher AST/ALT with trough concentrations > 5 mg/L. Neither dose nor trough alone correlated well with AUC, but AUC can be estimated with one sample 10 h after the first maintenance dose or a trough concentration, if combined with a Bayesian approach or a peak and trough without a Bayesian approach.

## 1. Introduction

Invasive fungal infections (IFIs) are a major source of morbidity and mortality in immunocompromised pediatric patients. The incidence of IFI in patients with acute leukemia or post stem cell transplant varied from 15–25% in one report [1]. Of these patients, estimates of mortality are wide but significant, with 10–70% dying from IFI, depending on the patient population. The most common causes of IFIs are *Candida* species, *Aspergillus* species, and organisms from the *Mucorales* family [2]. The primary therapy for invasive *Aspergillus* in children is considered to be voriconazole, which also has activity against *Candida* and many other fungi, but complex, highly variable pharmacokinetics and significant toxicities, plus lack of activity against *Mucorales*, make alternative therapy desirable [3]. Amphotericin B has a broader spectrum of activity than voriconazole, often used empirically for anti-fungal treatment or if voriconazole is not tolerated or active, but its significant nephrotoxicity and lack of oral formulation make Amphotericin B no longer the antifungal of choice for pediatric IFIs when alternatives are available.

Isavuconazole is a triazole that was FDA approved in 2015 and shown to be non-inferior to voriconazole in treating invasive aspergillosis [4]. Isavuconazole is the only other drug in addition to Amphotericin B approved by the US FDA to treat invasive mucormycosis. In adults, isavuconazole appears to be better tolerated than voriconazole, with reduced hepatobiliary, eye, and skin disorders [4]. While isavuconazole has been approved for use in adults by the FDA, and numerous case reports and series indicate that it is likely safe and effective in pediatric populations, it is not yet approved for use in children younger than 18 years of age.

Based on a prospective study of isavuconazole pharmacokinetics (PK) in pediatric patients receiving the drug for anti-fungal prophylaxis, a pediatric dosage regimen was proposed to match isavuconazole exposures after approved dosing in adults [5]. The proposed pediatric regimen begins with a loading dose of 10 mg/kg/dose IV/PO every 8 h on days 1 and 2, followed by 10 mg/kg once daily with a maximum dosage of 372 mg. Dosing is based on the pro-drug isavuconazonium, and 372 mg of isavuconazonium is equivalent to 200 mg of isavuconazole. We wished to verify this dosage regimen in another pediatric patient population and to explore pharmacokinetic/pharmacodynamic (PK/PD) relationships.

During the first year of isavuconazole use at our hospital, we made it standard practice to measure serum drug concentrations due to the lack of an approved pediatric dose. We now report the results of our retrospective review of those initial patients at our hospital who were prescribed oral and/or intravenous isavuconazole with available measured serum concentrations. Our goals were four-fold: (1) to summarize the usage in our population; (2) to estimate isavuconazole serum concentration-time profiles and characteristics using a population approach, comparing to adult exposures and the previous pediatric study [5]; (3) to assess for relationships between serum concentrations and both mortality and hepatotoxicity (PK/PD); and (4) to devise a maximally informative, optimized sampling regimen to measure serum isavuconazole in our pediatric patients.

## 2. Materials and Methods

### 2.1. Study Design

We conducted a retrospective review of patients at the Children’s Hospital Los Angeles (CHLA). The CHLA Institutional Review Board (IRB) approved the study and extraction of data between 1 January 2019 and 31 May 2020, the period during which we routinely collected blood samples for isavuconazole therapeutic drug monitoring (TDM). The IRB waived the requirement for informed consent since we only used existing clinical data, all of which was de-identified for this project.

### 2.2. Subjects

We searched our hospital electronic medical record system for eligible patients identified as those who met both of the following criteria: (1) received isavuconazole as part of their standard inpatient care within the study period, and (2) underwent inpatient TDM with measured serum drug concentrations of isavuconazole. We allowed inpatients who were continuing isavuconazole therapy from the outpatient setting, but we restricted our data extraction to hospital data because we could not verify outpatient isavuconazole dosing times.

### 2.3. Data Collection

We searched our hospital electronic medical record system for eligible patients identified as those who met both of the following criteria: (1) received isavuconazole as part of their standard inpatient care within the study period and (2) underwent inpatient TDM with measured serum drug concentrations of isavuconazole. We allowed inpatients who were continuing isavuconazole therapy from the outpatient setting, but we restricted our data extraction to hospital data because we could not verify outpatient isavuconazole dosing times.

For each included patient, we extracted demographic, dosing, concentration, and clinical data, which we stored in an Excel spreadsheet. The dosing data included isavuconazonium dose date and time, amount, and route of administration. For intravenous doses, the standard infusion time was one hour. The concentration data included date and time of the sample and the measured isavuconazole concentration. All blood samples were obtained for routine TDM. The CHLA isavuconazole TDM protocol was to obtain a trough concentration within one hour of the next dose. However, due to the long half-life of the drug, random times were acceptable if more convenient for patient management. Samples were to be obtained after maintenance dosing commenced. The exact number of maintenance doses prior to sampling varied but generally ranged between 3 and 7 unless the patient was already receiving isavuconazole when admitted. The therapeutic serum target in the first few days of dosing was >0.5 µg/mL, since steady state was not expected until at least one week. The ultimate steady-state concentration goal was >1 µg/mL based on the lower limit of the day 7 trough reported in the SECURE study (Supplemental Data) [4]. There was no established upper safety limit for the isavuconazole concentration.

For clinical data, we recorded underlying medical conditions, details of the fungal infection including diagnostic certainty using EORTC criteria [6], location and pathogen if identified, serum alanine transaminase (ALT), aspartate aminotransferase (AST) and mortality with attributable cause as abstracted from the medical record.

### 2.4. Isavuconazole Measurement

The CHLA clinical laboratory sent frozen serum to Eurofins Viracor for measurement of isavuconazole. The turn-around time was 3–5 days. Eurofins Viracor used methanol protein precipitation and a validated high-performance liquid chromatography/tandem mass spectrometry (HPLC/MS/MS) assay with deuterated isavuconazole as an internal standard. The assay linear range was 0.10–10.0 µg/mL, with intra-assay imprecision of ≤3%, and inter-assay imprecision of ≤4%. The lower limit of detection was 0.041 µg/mL. The bias of low, middle and high control samples ranged from −3% to 0%.

### 2.5. Population Modeling

For all data analysis and figures, we used R (4.2.1, R Core Team, Vienna, Austria), including for population modeling and simulation with the Pmetrics package [7]. To pre-process the data, we first ordered doses and concentrations by date/time for each subject. When no documented loading doses were available for a given subject and hospital admission, we assumed that the isavuconazole was carried forward from outpatient dosing prior to admission and added 6 loading doses and 7 daily maintenance doses to the beginning of the record to account for concentrations measured during the first week of admission. For these prepended doses, we used the same dose amount as recorded on admission in the medical record. We included all admissions for subjects with multiple hospitalizations within the study period. Because sampling was typically very sparse, we did not attempt to estimate interoccasion variability in model parameter values. We also recorded measured weight, AST and ALT and ordered them with respect to date/time obtained and dates/times of isavuconazole doses and concentrations. We assumed weight, AST and ALT to change linearly between measurements for a given patient. We assumed height to be unchanged during each admission.

Within Pmetrics, we used the non-parametric adaptive grid algorithm [8]. We tested increasingly complex models, starting with the simplest model comprising linear absorption (Ka) from an oral bolus compartment into a central compartment with volume V, clearance (CL), intercompartmental clearance (Q) and volume of the peripheral compartment (Vp). After finding the model with a peripheral compartment to be superior, we compared it to a model parameterized with elimination, Ke, and transfer to (KCP) and from (KPC) a peripheral compartment. For both CL and Ke models, we tested inclusion of weight normalized to a reference weight of 70 kg as an allometric multiplier, using an exponent of −0.25 for rate constants, 0.75 for clearance terms and 1.0 for volume terms. In addition to these fixed exponents, we compared models with fitted allometric exponents and age-dependent exponents as suggested by Mahmood et al. [9].

For the prediction error model, NPAG uses fixed coefficients that define a polynomial equation, which in turn defines the standard deviation of any observation, such that $SD=C_{0}+C_{1}*\left[c\right]+C_{2}*[c{]}^{2}+C_{3}*[c{]}^{3}$, where $\left\{C_{0},\ldots ,C_{3}\right\}$ are the fixed coefficients and $\left[c\right]$ is the observed drug concentration. We set $C_{0}=C_{1}=0.1$, and $C_{2}=C_{3}=0$, which is an additive and proportional error model related to the measurement. Ideally, coefficient values are determined by fitting the polynomial equation to assay validation data, but the values we chose are typical when assay validation data are unavailable. We used lambda (λ) to capture additional noise related to uncertainty in the data (e.g., small misspecifications in dose amount or dose/sample times) and model misspecification, such that individual observations were fitted by NPAG with weights of the inverse of the variance, i.e., ${\left(\lambda +SD\right)}^{-2}$. λ was optimized over the population.

For model selection, we were particularly interested in mean posterior prediction bias (mean weighted prediction minus observation) to evaluate the models, since we were most interested in estimating individual exposures to correlate with clinical outcomes. However, because we also wished to use the model for simulation, we evaluated the Akaike Information Criterion, population bias, population/posterior imprecision (bias-adjusted, mean weighted squared prediction minus observation) and the normalized prediction distribution error (npde) method of Comets et al. [10], using the npde package (version 3.2) for R, to which Pmetrics links. Overall, we chose the model with the best performance in the largest number of these criteria.

### 2.6. Use of the Population Model

We had four major objectives for the accepted model: (1) to compare the exposures in our pediatric patients to reference adult exposures reported in the package insert after approved dosing and to previously reported pediatric exposures [5]; (2) to compare simulated pediatric exposures associated with the proposed dosing of 10 mg/kg every 8 h for 2 days, followed by once daily thereafter to the reference exposures; (3) to test whether average daily trough and/or 24-h area under the time-concentration profile (AUC) were associated either with mortality or with ALT/AST as markers of toxicity; and (4) to calculate the optimal time to sample for isavuconazole measurement to best estimate AUC.

For objective 1, since published reference adult values in the package insert are at steady state, we appended an additional 20 regularly timed doses administered every 24 h to the end of each patient’s data file. This permitted us to analyze their individually predicted concentration-time profiles at steady state. We used the final dose and subsequent 24-h concentration profile for these calculations. We extracted each subject’s projected steady-state maximum concentration (Cmax), time to Cmax (Tmax) for oral dosing and minimum (trough) concentration. We conducted non-compartmental analyses including estimation of AUC by the trapezoidal approximation and calculation of half-life (T-half) by linear regression on the final 5 predicted concentrations. We calculated clearance by the equation CL = Dose/AUC, assuming 100% bioavailability for oral doses (reported as 98% in the Cresemba package insert). However, we did adjust the dose by a factor of 0.54 for this calculation of CL to reflect that isavuconazole is supplied as the prodrug isavuconazonium, for which 372 mg = 200 mg of active compound and 200/372 = 0.54.

For objective 2, we used the simulation function in Pmetrics and the population model to estimate the exposure after 3 weeks of daily dosing following an initial two-day loading with doses every 8 h on each of the first two days (six total loading doses). All simulated doses were 10 mg/kg. Of the 2000 simulated patients, half were oral dosing and half were intravenous. We used the mean weight in our study population (50 kg) and the SD of the weight (27 kg) for the purposes of simulating the weights. We used a mean age of 10 and SD of 8 years, limiting the range to <18 years. Pmetrics maintains all population correlations between covariates and between covariates and parameters when simulating. As above, we calculated Cmax, Tmax, trough concentration, AUC, T-half and CL from the final 24 h of each simulated profile.

For objective 3, we calculated the average daily trough concentration and AUC from each subject’s individual, Bayesian-posterior predicted concentration-time profiles, with predictions at one-hour intervals. To estimate only the actual subject exposures, we omitted the trailing 20 doses we had appended to predict steady state PK profiles for objective 1 above. For the average daily AUC, we calculated the total AUC for the entire study period for each subject, divided this number by the total time in hours and multiplied by 24 to obtain the daily average. Then, by Wilcoxon rank sum for univariate testing or multiple linear regression, we tested the associations between isavuconazole trough concentration and AUC (log transformed to satisfy assumptions of normality) vs. ALT and AST AUC, calculated in a similar manner as for isavuconazole AUC but only dividing total AST/ALT AUC by total time to generate an estimate of average ALT/AST over the study period. By multiple logistic regression, we tested the association between isavuconazole trough concentration, AUC and mortality, both crude and attributable to the IFI. For all regressions, we additionally included dose, weight, height, age and sex as possible predictors. We used the “step” function in R to test predictors by forward and backward elimination.

For Objective 4, we used our multiple-model optimal (MMopt) sampling algorithm [11]. MMopt finds any number of times at which point the possible concentration time profiles arising from the discrete support points in the population model are the most separate for a given dosage regimen. This minimizes the risk of choosing a misinformative Bayesian posterior parameter value distribution for an individual by maximizing the information content of a given sample. We used the same regimen as for Objective 2. MMopt has a number of advantages over traditional D-optimal sampling [12], in particular, that the number of samples does not have to be equal to the number of model parameters, although the risk of choosing a misinformative prior drops with increasing sample number. MMopt can also be tuned or “weighted” to maximize discrimination of a particular statistic, and in this case, we chose to minimize the risk of estimating an incorrect AUC as our statistic of choice. Because we were interested in informing clinical practice, we tested only a single MMopt sample obtained in the 24-h window after the first maintenance dose from 48 to 72 h after start of therapy. The endpoint was predicted AUC from 48 to 96 h based on simulated concentrations every hour (full AUC) compared to the Bayesian posterior AUC estimates from simulated patients with only the MMopt sample available (i.e., one isavuconazole concentration or limited AUC).

## 3. Results

### 3.1. Study Population

We identified 26 patients who met the eligibility criteria, of whom 19 (73%) were male. The mean (SD) age was 12.7 (5.5) years with a range of 10.8 months to 20.3 years. Mean weight was 50.9 (26.8) kg, range 9.3 to 116.6 kg. Of the 26, 18 (69%) had pulmonary disease. The rest of the subjects suffered from either sinus or disseminated infections. Based on EORTC criteria, 10 (38%) subjects had proven disease: five were *Candida* species (one each for albicans, tropicalis, krusei, glabrata, and parapsilosis), one *Rhizomucor pusillus*, one *Absidia* sp., one mixed *Aspergillus fumigatus* and *A. terreus* and two *Mucorales* sp. Based on positive galactomannan with compatible radiographic findings in an immunocompromised host, an additional 8 (31%) had probable disease. The final 8 (31%) had possible disease. The underlying diagnosis was leukemia or lymphoma for 22 (79%) of the patients.

### 3.2. Therapeutic Characteristics

There were 8 episodes in 7 patients that required pre-pended doses due to outpatient dosing prior to admission. Only one patient had pre-pended doses for the first episode, i.e., the other 6 patients started isavuconazole in the hospital during a study period, were discharged and had at least one subsequent occasion/readmission while still taking isavuconazole. A median of 12 isavuconazole doses were modeled per subject, ranging from 6 to 65. Of 455 isavuconazole doses in the data, 152 (33%) were oral. For both routes of administration, 176 (39%) doses were capped at the adult dose of 372 mg in 17 (65%) of the 26 subjects. Among the subjects who received doses less than 372 mg, the mean dose was 9.8 mg/kg, with 80% between 9.7 and 10.6 mg/kg and a range between 5.9 and 11.3 mg/kg.

One sample was obtained from 18 of the subjects, two samples from three subjects, three samples from two subjects, and four to six samples from one subject each. The median time of sampling was 17.9 h after the previous dose, with 29 (64%) of the 45 samples as trough concentrations obtained between 20 and 25 h post-dose. The full distribution of post-dose sample times is shown in Figure 1. The median trough concentration was 3.0 mg/L, ranging between 0.9 and 10.0 mg/L. The median concentration obtained within five hours after the dose (roughly a peak for oral) was 3.8 mg/L, range 3.1 to 5.6 mg/L. Note that the median peak and trough were similar, reflecting the very long half-life. Furthermore, most patients did not have early concentrations, so the range of trough concentrations was greater than the peak.

### 3.3. Model Fit

Starting with models parameterized with clearance, the best model had orally administered drug transferring from the depot compartment with a fractional absorption rate constant (Ka) to the central compartment with volume V, clearance (CL) and distribution to and from (Q), a peripheral compartment with volume Vp. Models with delayed or fixed absorption or bioavailability increased the prediction bias and decreased the likelihood, so we discarded them. Fitting the allometric exponent for weight normalized to 70 kg on CL resulted in a mean value of 0.72, very close to the theoretical value of 0.75. The age-dependent fixed exponents were not better than the theoretical value of 0.75. Therefore, we retained the constant 0.75 model. However, re-parameterizing the CL model with rate constants resulted in improved likelihood and reduced prediction bias. Therefore, our final model had drug eliminated from the central compartment (Ke) and transferred to (KCP) and from (KPC) the peripheral compartment. Weight was normalized to 70 kg and scaled to each transfer rate with a fixed exponent of −0.25. Normalized weight was scaled to V by a fixed exponent of 1.

Summaries of the final model parameters are shown in Table 1. The discrete marginal distributions and final support points for the model parameters are shown in Appendix A, and observed vs. predicted plots are shown in Figure 2. The npde plot is shown in Appendix A.

Overall, while there was considerable variability in the parameter values, based on an excellent R^2^ for the posterior predictions with low bias and regression slope of close to one for both population and posterior predictions, npde in good agreement with expected distribution and moderate parameter shrinkage, we felt confident in the model to estimate individual exposures and to simulate expected exposures.

### 3.4. Objective 1: Steady State Pediatric Exposures Compared to Adult Benchmarks

Projected steady-state non-compartmental PK parameters in our population, given their individual dosing and body weight, and a comparison with adult values are shown in Table 2.

Overall, there was good agreement between pediatric and adult exposures, as summarized by Cmax, Tmax and AUC, with slightly higher median Cmax and AUC in adults. The half-life appeared to be somewhat shorter in children, but clearance was lower, implying by the relationship $t_{\mathrm{half}}=\frac{\ln\left(2\right)}{CL}*V$ that volume of distribution was smaller in the children. Indeed, we estimated median volume at steady state (V_ss_) to be 115 L, compared to a mean V_ss_ of 450 L in adults reported in the Cresemba package insert.

The full distributions of projected steady-state trough concentrations and AUCs are shown in Figure 3. Overall, the trough concentrations and the AUCs for children were within adult ranges, but on the lower end. Figure 4 shows the lack of relationship between isavuconazonium dose in mg/kg and average daily AUC and a marginal linear relationship between isavuconazole trough concentration and AUC. The variability around the linear regression between trough concentration and AUC was wide, such that the AUC 95% prediction interval for the median study population trough concentration of 3 mg/L was 77 to 163 mg∗h/L, more than a two-fold difference.

### 3.5. Objective 2: Simulated Pediatric Exposures from 10 mg/kg Dosing Compared to Adult Benchmarks

Table 2 includes a comparison between adult values compared with distributions in 2000 simulated pediatric patients. Dosing for the simulated patients was 10 mg/kg IV or oral, 3 times daily for two days, followed by a week of once-daily dosing. Doses were capped at the adult dose of 372 mg in accordance with the proposed regimen [5] and likely typical clinical practice. Median (range) simulated weights and ages were 47.3 (10.6–77.3) kg and 10.2 (2.2–17.2) years, with the same correlation as the study population between weight and age and between these covariates and the model parameters. As for the projected steady state metrics in our study population, overall, there was good agreement between simulated pediatric and adult exposures. Again, half-life was shorter in children.

The distributions of simulated steady-state trough concentration and AUC are shown in Figure 5. Overall, the simulated steady-state trough concentrations and AUCs for children were within adult ranges but on the lower end.

### 3.6. Objective 3: PK/PD Relationships

#### 3.6.1. Mortality

Of the 11 deaths in the study population, five could be attributed at least partially to their IFI (3 proven, 2 probable), and for the remaining six, it was uncertain if the IFI (2 proven, 2 probable, 2 possible) contributed to mortality. Among all patients who died (crude mortality), there was no clear relationship between death and average daily isavuconazole trough, AUC, or AUC > 91.58 mg∗h/L, which was the population median, as shown in Table 3, top half.

However, when considering mortality attributed to the IFI vs. those who survived or died from causes other than an IFI, average daily AUC was lower in those with attributed mortality, and all the patients who succumbed to their IFI had an average daily AUC < 91.58 mg∗h/L (Table 3, bottom half). By univariate analysis, median AUC was higher in those without attributed mortality vs. those with attributed mortality (104.2 vs. 53.7, *p* = 0.034, Wilcoxon rank sum), as shown in Figure 6. Further multivariate logistic regression analysis with forward/backward elimination to predict attributed mortality identified only average daily AUC (*p* = 0.058) as a borderline significant predictor controlling for weight, age, trough concentration and dose. Although trough concentration was not a significant predictor of attributable mortality, average daily steady-state trough concentration did tend to be lower in this group. All of the four patients with a steady state daily AUC of <60 mg∗h/L (the lower threshold used by Arietta et al. [5]) had average daily steady-state trough concentrations < 2 mg/L, although 2 of the 5 with attributed mortality were above this trough threshold.

#### 3.6.2. Hepatotoxicity

Average AST and ALT were commonly elevated above 40 IU (AST: *n* = 14, 54%; ALT: *n* = 15, 58%) in this population with polypharmacy and multiple co-morbid conditions. The maximum values observed were 1144 IU and 698 IU for AST and ALT, respectively. We defined emergent hepatoxicity as either an increase to >3× upper limit normal (ULN) for those with normal baseline AST/ALT, or 3× baseline for those with baseline abnormal enzymes (>1× ULN). For both AST and ALT, 14 (54%) of the patients had elevated values at baseline. Four (15%) had emergent toxicity by AST and ALT and an additional patient by ALT only (5 total, 19%) while taking isavuconazole. Although the ALT was improving in 3 (60%) of the 5, it had not returned to baseline by the end of the study period in any subject.

By multivariate linear regression with forward and backward elimination, a mean daily isavuconazole trough > 5 mg/L was associated with higher average daily AST (*p* = 0.003) and ALT (*p* = 0.005), controlling for mean daily isavuconazole AUC and dose. By multiple logistic regression, controlling for AUC and dose, a trough of >5 mg/L was also independently associated with occurrence of treatment-emergent elevations for AST (*p* = 0.03) and for ALT (*p* = 0.05), as shown in Figure 7.

### 3.7. Objective 4: Optimal Sampling

We restricted the optimal sampling window to between 48 and 72 h after start of therapy to allow for completion of all 6 loading doses and the first maintenance dose yet balance the need to assess exposure early. The single optimal sampling time for both IV and oral dosing was 58 h, i.e., 10 h after the first maintenance dose. We compared the “Full” simulated AUC from 48 to 96 h to the “Limited” AUCs calculated from the median of the Bayesian posterior parameter distributions based on the single MMopt sample in each of 200 simulated subjects (half IV dosing, half oral dosing) or based on a trough. The distributions of full vs. limited AUCs are shown in Figure 8. For IV administration, the geometric mean full AUC was 248; for MM, it was 244; and for trough, it was 229 mg∗h/L (*p* = 0.45, analysis of variance). For oral administration, full AUC was 225, MM was 237 and trough was 253 (*p* = 0.14).

## 4. Discussion

In the most comprehensive report thus far of isavuconazole PK/PD in children with possible, probable or proven invasive fungal infections (IFI), we found that the proposed dose of isavuconazonium 10 mg/kg, capped at the adult dose of 372 mg, is likely safe and effective in children, at least as an initial regimen. Our study population, who largely received this dose or close to it, had projected average steady-state trough concentrations and AUCs very similar to previously studied children given the same dose for prophylaxis [5], who in turn were very similar to adults with IFIs who were treated with isavuconazole [4]. With our population model developed from the real subjects, we demonstrated similar average exposures in a simulated population of children ranging from 2 to 18 years and 10 to 77 kg dosed according to the proposed regimen above. Nevertheless, there was considerable interpatient variability in isavuconazole concentrations.

For PK/PD relationships, we found that all 5 subjects whose deaths were attributable to their IFI had serum isavuconazole average daily AUCs below 90 mg∗h/L, and 4 of the 5 were below 60 mg∗h/L, which was a proposed lower limit in a previous pediatric study of isavuconazole PK used for prophylaxis [5]. Although the median daily trough concentration was also lower in those with IFI-attributed mortality, there was substantial heterogeneity, making it difficult to establish a threshold: three were below 2 mg/L, one was between 2 and 3 mg/L, and one was >5 mg/L on average.

A question arises whether isavuconazole trough concentrations are generally a good surrogate for AUC when AUC is the true isavuconazole target based on murine studies of invasive candidiasis [13] and aspergillosis [14], consistent with animal studies of other azoles, as well as our own findings regarding attributable mortality. We found that while there may have been a statistically significant linear relationship between isavuconazole trough concentration and AUC, the variability around the regression line, with an R^2^ value of only 0.13, forces us to conclude that isavuconazole trough concentrations do not predict AUC with enough precision to truly ensure the AUCs greater than those we found to be associated with attributed mortality. This is the same as we have found for vancomycin [15,16,17] and may be part of the reason why there is no clearly established trough concentration target for isavuconazole. However, we did find that when combined with our population model, a limited trough sampling strategy after the first maintenance dose predicts AUC with an accuracy that cannot be statistically distinguished from an optimally timed sample 10 h after the same dose. The similarity between trough and optimal sample is likely due to the long half-life of the drug. Nevertheless, the trough concentration only became a useful predictor of AUC when combined with a population model.

In contrast to attributable mortality, we found that the measured trough concentration was more predictive of increases in liver enzymes than AUC. There was a highly significant linear relationship between average trough concentration > 5 mg/L and either higher average AST or ALT and a borderline association with average trough concentration > 5 mg/L and emergent hepatotoxicity on treatment. Since we analyzed average trough concentration, this threshold is not meant to be one that is never exceeded but merely a suggestion that prolonged trough concentrations above this will likely increase the risk of developing some degree of enzyme elevation. As we found for voriconazole [18], many children will experience bumps in their enzymes while on azole therapy, but will regress towards or achieve baseline values without the need to stop or change therapy.

A previous secondary analysis of the relationship between isavuconazole exposure metrics and outcomes including mortality and liver enzyme elevations in adults from the SECURE study found no discernible PK/PD relationship [19]. While it is possible that the relationships we found were by chance, despite the *p*-values, given our very small population size relative to the SECURE study, there are some differences worth noting. First, we examined IFI-attributed mortality, which we believe to be more relevant in a highly co-morbid population to antifungal PK/PD rather than crude mortality as measured by Desai et al. Second, we had a higher level of liver enzyme elevation in approximately 50% of our subjects compared to their surprisingly low ~10% rate. We also estimated average AST/ALT as the area under the total concentration time profile divided by the total time, which corrects for sparse or irregular sampling, and used that as our outcome, rather than individual values. Third, the isavuconazole exposure variability (CV%) in the adult study was reported as 62%, while it was 92% based on average daily AUC in our study. This equated to a 25-fold AUC difference between the highest and lowest subjects, and the broader range may have allowed us to find PK/PD relationships not previously demonstrated. This last point has important therapeutic implications. Despite a 4.6-fold difference in dose (mg/kg) across our pediatric population, the AUC difference was much greater, and there was no correlation between dose and AUC. Clearly, the same dose, even adjusted for bodyweight, does not lead to predictable or consistent exposures in a real-world population of children with IFI.

Regarding our model, we found the best to be a central compartment with a single peripheral compartment. The prior pediatric population PK model found an additional peripheral compartment improved the predictions [5]. However, this is likely due to more intensive sampling than was available to us, and our model did not appear to suffer in its ability to predict exposures very closely in line with prior reports. On a final note about our model, there is great debate in the pharmacometric literature on whether allometric exponents should be fixed, fixed with a maturation function, fixed according to age, or fitted. We found that fitting the exponent on clearance resulted in an allometric exponent of 0.72, very close to the theoretical value of 0.75, and that age-dependent fixed values were not any better. Isavuconazole is highly bound to albumin, and it is metabolized extensively by CYP3A4/5. For such drugs, where average CYP3A4/5 enzyme maturation is 76% of adult values by one year of age and 92% by 5 years of age (van Rongen et al., their Table 4) [20], allometry using fixed exponents without a maturation function is expected to be the preferred method of scaling (van Rongen et al., their Table 2) [20]. This is exactly what we found in our population, which had 1 subject 10.8 months, 1 subject 2.6 years old, 1 subject 3.3 years old, and the remaining 23 subjects aged 5 years and older.

The main limitation of our study is the small sample size. It was also retrospective, which means that there can be confounding factors that we were not able to control that affected outcomes, such as other contributions to mortality or to hepatoxicity. Nevertheless, despite these limitations, we believe we have shed some light on the PK/PD of isavuconazole in the pediatric population. We found that an average daily AUC of <60 mg∗h/L was associated with increased risk of mortality attributed to IFI. We also found that risk of hepatotoxicity increased with mounting isavuconazole trough concentrations and that 5 mg/L may be a reasonable threshold above which extra caution is warranted.

Finally, unlike previous conclusions in adults [19], we feel that the poor correlation between dose and serum concentrations, the wide variability of serum exposures in children, and the suggestion of exposure response relationships all merit strong consideration of optimized dosing based on measured drug concentrations in these incredibly vulnerable patients with little room for therapeutic error. As for vancomycin, AUC-guided dosing appears best for efficacy, with trough-guided dosing to avoid hepatotoxicity. Obtaining a steady-state peak concentration in addition to the trough concentration would permit estimation of AUC by fundamental PK equations, as we have outlined for vancomycin [21]. The disadvantage of this strategy is the need to wait for steady state, which is likely not until after the 5th maintenance dose (i.e., day 7 or beyond). A single steady-state trough sample of >2 mg/L may predict an AUC above a proposed lower threshold of 60 mg∗h/L [5], but we did not find either threshold to be predictive of survival. In our opinion, a better approach is to obtain either one sample 10 h after the first maintenance dose or a trough concentration combined with a Bayesian algorithm and a population model to estimate AUC, targeting a steady-state value > 100 mg∗h/L.

## Figures and Tables

**Figure 1 pharmaceutics-15-00075-f001:**
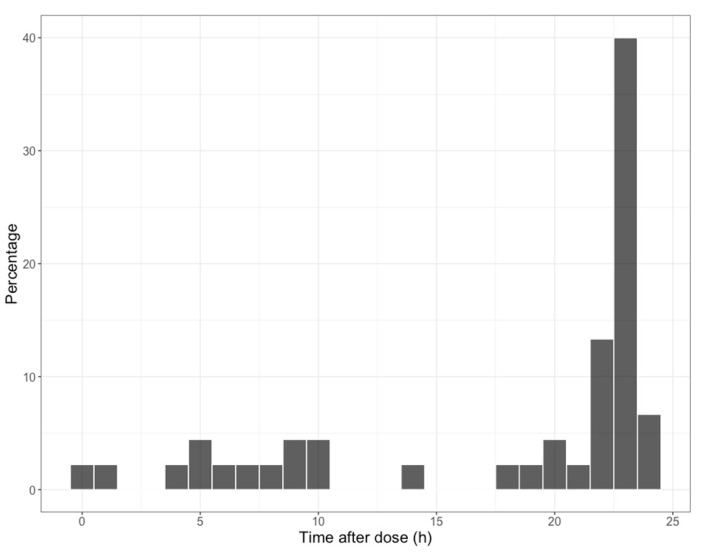
Distribution of blood sampling times after prior isavuconazole dose in the population. The majority were trough concentrations near the end of the dosing interval (24 h).

**Figure 2 pharmaceutics-15-00075-f002:**
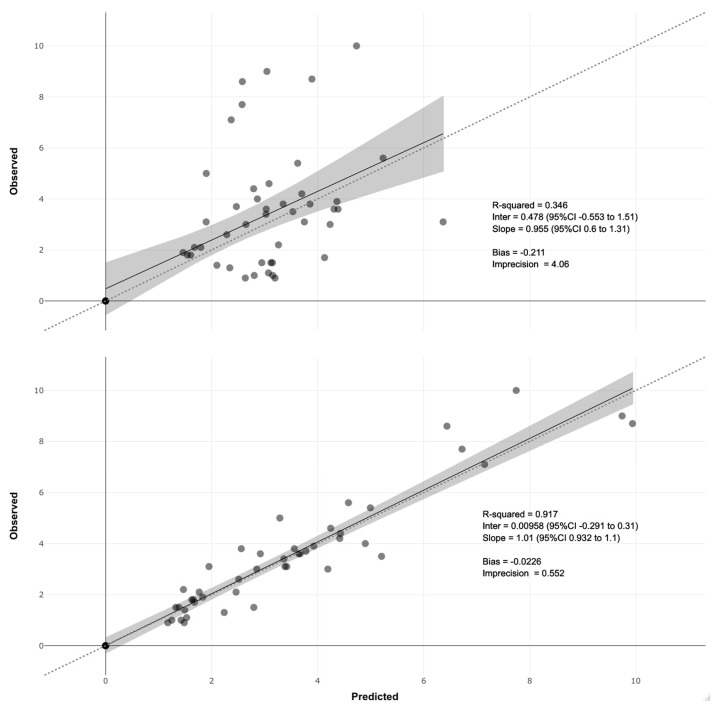
Observed vs. predicted plots. Predictions are based on population median parameter values in the top plot and posterior medians in the bottom plot. Bias is mean weighted prediction error, and imprecision is mean, bias-adjusted, weighted squared error. Dotted lines are identity with slope of 1 and intercept of 0. Shaded regions are 95% CI around the solid mean regression lines.

**Figure 3 pharmaceutics-15-00075-f003:**
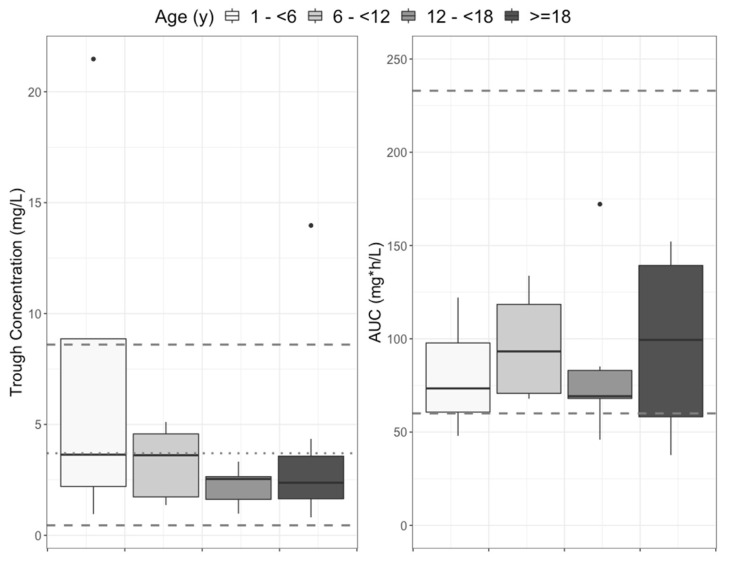
Distribution of projected steady-state trough concentrations (**left**) and 24-h AUC (**right**) in the current study population. For trough concentrations, dashed lines are the ranges (0.45 and 8.6 mg/L) and the dotted line is the median (3.7 mg/L) of the assumed steady state for adults, based on concentrations measured on day 14 in the SECURE study (included in their published Supplemental Data) [4]. For AUCs, the upper dashed line (233 mg∗h/L) is the minimum adult AUC in a high-dose isavuconazole study with increased toxicity, used as an upper threshold by Arietta et al. [5]. The lower dashed line (60 mg∗h/L) is the 25th percentile for adults in the SECURE study, again used by Arrieta et al. as the lower target in their pediatric study. The horizontal solid line in each box is the median, the upper and lower box margins are the interquartile range (IQR, 25th and 75th percentiles), the whiskers extend to 1.5× IQR, and points are outliers.

**Figure 4 pharmaceutics-15-00075-f004:**
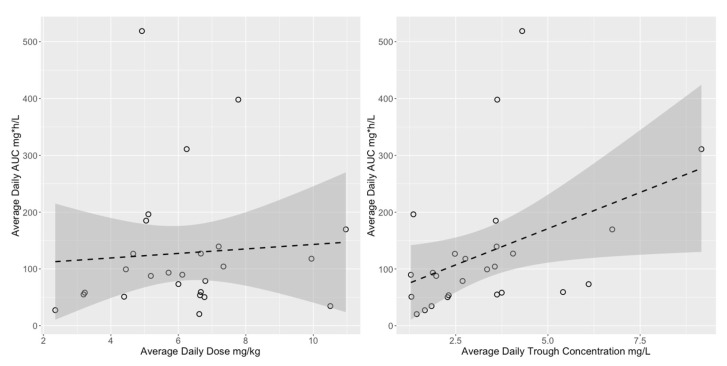
Relationship between average daily isavuconazole AUC and dose (**left**) or trough concentration (**right**). Dashed line is the linear regression and shaded area is the 95% confidence interval. Regression statistics for the dose were slope = 0.35, *p* = 0.73, R^2^ = 0.004. For the trough concentration they were slope = 25.5, *p* = 0.04, R^2^ = 0.13.

**Figure 5 pharmaceutics-15-00075-f005:**
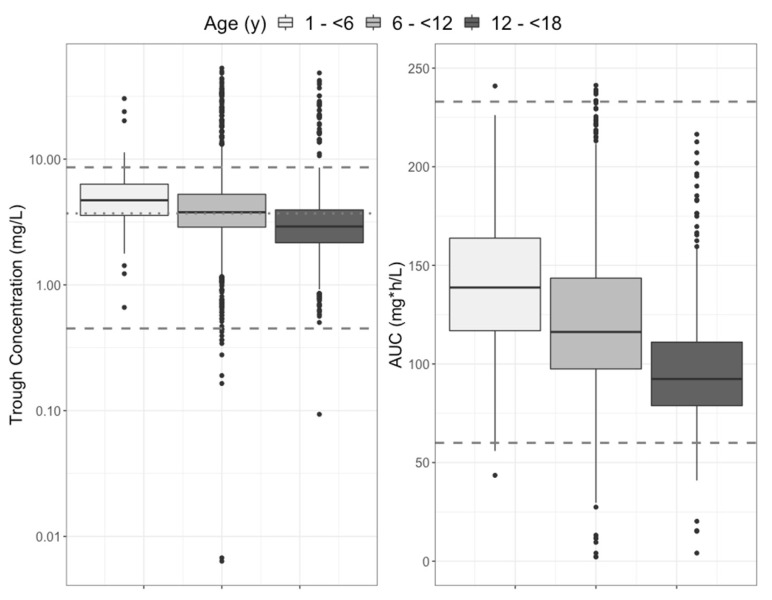
Distribution of steady-state trough concentrations (**left**) and 24-h AUC (**right**) in 1000 simulated patients administered isavuconazole 10 mg/kg/dose IV q8h on the first two days and then daily for a week. Reference lines are the same as for Error! Reference source not found.

**Figure 6 pharmaceutics-15-00075-f006:**
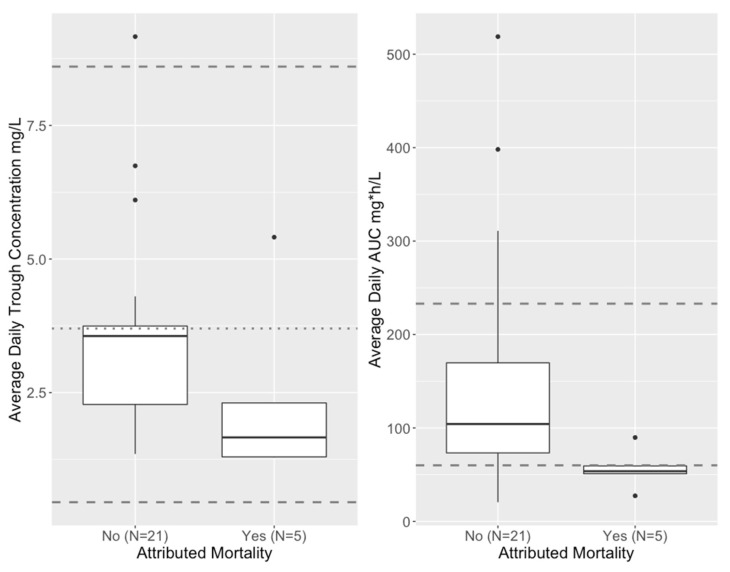
Distribution of isavuconazole average daily trough concentrations (**left**) and AUCs (**right**) in subjects who either survived or were not considered to have died from their IFI (attributed mortality = NO) vs. those who were considered to have died from their IFI (Attributed Mortality = YES). The same adult reference concentrations as for Table 2 are indicated by horizontal dashed lines. The median trough concentrations were 3.6 vs. 1.6, *p* = 0.09 (Wilcoxon rank sum) and median AUCs were 104.2 vs. 53.7, *p* = 0.03, Wilcoxon rank sum.

**Figure 7 pharmaceutics-15-00075-f007:**
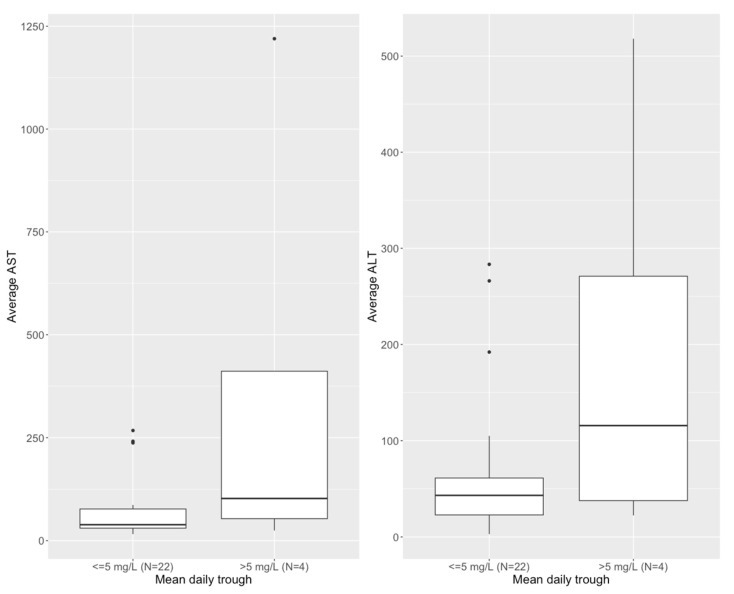
Distribution of average daily AST (**left**) and ALT (**right**) in study subjects, stratified by mean daily isavuconazole trough concentration.

**Figure 8 pharmaceutics-15-00075-f008:**
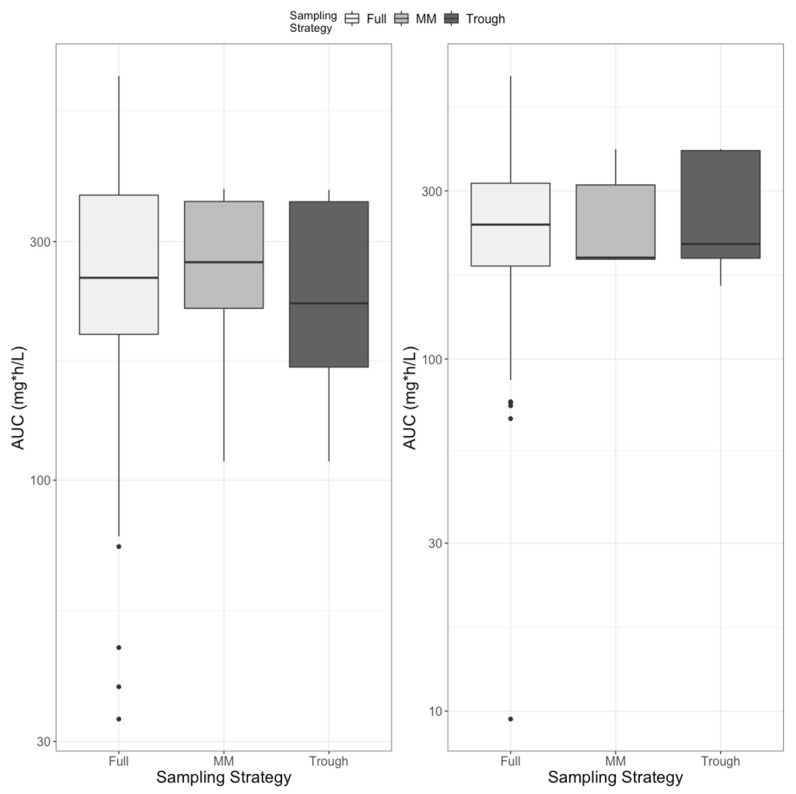
Distribution of AUC for IV (**left**) and oral (**right**) dosing comparing values calculated from concentrations simulated every hour (Full) or predictions from models using only a single MMopt sample (MM) at 10 h after the first maintenance dose or the trough concentration (Trough) after the first maintenance dose to generate the Bayesian posterior parameter value probability distributions.

**Table 1 pharmaceutics-15-00075-t001:** Population parameter value summaries. Ka, absorption; KCP0, distribution from central to peripheral compartment; Ke0, elimination from central compartment; KPC0, distribution from peripheral to central compartment; V0, volume of the central compartment.

Parameter	Median (95% CI)	Range	Shrinkage
Ka (h^−1^)	12.00 (0.03–12.00)	0.01–12.00	48%
KCP0 (h^−1^)	2.42 (0.14–4.39)	0.10–6.38	41%
Ke0 (h^−1^)	0.10 (0.05–0.12)	1.52 × 10^−05^–0.12	40%
KPC0 (h^−1^)	0.51 (0.13–1.34)	0.03–5.00	57%
V0 (L)	45.50 (37.27–53.92)	24.58–73.32	45%

**Table 2 pharmaceutics-15-00075-t002:** Comparison between steady-state pharmacokinetic characteristics in the current study population with reference values, arranged by dosing route (IV with white and Oral with grey background). Letters correspond to Objective, but rows are arranged to facilitate comparison between individual steady state (Objective 1), simulated steady state (Objective 2), and a previously reported external dataset of pediatric patients [5]. All rows are pediatric except for Row G. Rows A, B: Projected steady state in the current study population. Except for Row G, all data are presented as median (95% ile). Rows C, D: Steady state in simulated pediatric patients with 10 mg/kg loading every 8 h on days 1 and 2, followed by a week of daily dosing. Rows E, F: Weighted median (range) in pediatric patients extracted from Table 2 in Arrieta et al. [5] by taking the weighted median across age groups. Row G: geometric mean values from the Cresemba package insert.

	Population (N)	Route	Cmax (mg/L)	Tmax (h)	Thalf (h)	AUC (mg∗h/L)	CL (L/h)
A	Current (26) (Objective 1)	IV	5.9 (4.2–18.6)	–	49.5 (8.8–2767)	70.8 (41.4–336.0)	1.6 (0.3–4.8)
C	Simulated (1000) (Objective 2)	IV	8.8 (5.8–29.1)	–	42.1 (7.9–1959.3)	115.9 (65.0–639.6)	1.7 (0.3–3.1)
E	External (45)	IV	6.6 (3.44–9.96)	–	NR	86.3 (43.0–179.0)	NR
B	Current (26) (Objective 1)	Oral	6.0 (4.0–14.6)	2.0 (0–9)	44.7 (8.7–3183)	107.3 (70.1–295.0)	1.4 (0.5–2.3)
D	Simulated (1000) (Objective 2)	Oral	6.8 (4.0–22.8)	1.0 (1.0–7.0)	37.4 (7.8–1857.2)	112.8 (62.0–532.8)	1.7 (0.4–3.2)
F	External (45)	Oral	5.6 (2.0–8.9)	4.0 (2.0–8.0)	NR	97.7 (37.6–185)	NR
G	Adult (37)	Oral	7.4	3	130	121	2

**Table 3 pharmaceutics-15-00075-t003:** Relationship between isavuconazole exposures and survival outcomes. Trough and AUC are within-patient average daily values. The AUC threshold of 91.58 mg∗h/L is the population median. Mortality is considered in two ways: crude (top section), and mortality attributed by treating clinicians to the invasive fungal infection (bottom section).

	Survived (*n* = 15)	Died (*n* = 11)	*p*-Value (Test)
Trough (mg/L)	3.6 (1.9–6.5)	2.3 (1.2–8.2)	0.10 (Wilcox)
AUC (mg∗h/L)	93.4 (40.1–476.6)	89.8 (22.2–282.3)	0.38 (Wilcox)
AUC > 91.58	8 (53%)	5 (45%)	1 (Fisher)
	Non-attributed mortality or survived (*n* = 21)	Attributed mortality (*n* = 5)	
Trough	3.6 (1.4–7.9)	1.7 (1.3–5.1)	0.09 (Wilcox)
AUC	104.2 (27.5–458.5)	53.7 (29.7–86.7)	0.03 (Wilcox)
AUC > 91.58	13 (62%)	0 (0%)	0.04 (Fisher)

## Data Availability

De-identified data are available by reasonable request to corresponding author.

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
