# Peer review of "Isavuconazole Pharmacokinetics and Pharmacodynamics in Children"

_pharmaceutics, 2022, doi:10.3390/pharmaceutics15010075_

Round 1

Reviewer 1 Report

About this study

Although is not about a great number of subjects (only 26 patients) this study is interesting and could be of help to those interested in the nonparametric population models and their efficiency.

Introduction: is frugal even "spartan" but introduces the necessary elements to understand the necessity of this study. Even though, please add more data from the main-stream, about why is necessary (or not) to introduce isavuconazole in pediatric patients, knowing the potential toxicity of this triazole compound (here the risk of hepatotoxicity is increasing with mounting isavuconazole trough concentration, 5 mg/L being a limitation in this case).

M&M: is well written. The "Population modeling and Use" are following all four objectives and are enough clear to be replicable. Please add for HPLC/MS/MS the producer's name for the used analytic system.

Results and Discussion are the best part of this manuscript. Data are abundantly presented (including here the supplemental tables and figures) and associated with specific statistics/objectives and then commented on accordingly.

References: in general are well chosen and of actuality from the last decade.

Decision: Minor correction 

Reviewer 2 Report

This manuscript is exhibiting a retrospective pharmacokinetics (PK) and pharmacodynamics (PD) study for the broad-spectrum anti-fungal azole Isavconazole (Isa) in 26 pediatric patients (with probable Candida and Aspergillus spp disease) who received Isa as part of their standard inpatient care within the study period or underwent inpatient therapeutic drug monitoring with measured serum drug concentrations of Isa. The authors constructed a nonparametric population model using Pmetrics to analyze the extracted demographic, dosing, concentration, mortality, and hepatoxicity data. They found that the predicted steady-state Isa concentrations in patients were similar to previous reports in children and adults and simulations with the proposed dosing of 10 mg/kg/dose every 8 hours for 2 days followed by once daily maintenance matched effective adult exposures. They further found that the attributable mortality was significantly associated with steady-state daily AUC less than 60 mg*h/L and higher AST/ALT with trough concentrations higher than 5 mg/L. 

Overall, this is a well-described and interesting manuscript. Some suggestions for the authors may make the manuscript more understandable to general readers.

1. Apparently, there are some problems with text editing in which all the cited figures are not depicted in the text but only show "Error! Reference source not found.". Please revise them accordingly.

2. The descriptions for the populations as indicated with A~G are very confusing in Table 1. Separation lines to properly separate these populations would make a significant improvement.

Reviewer 3 Report

The manuscript describes the results of population and statistical analyses of data collected from pediatric patients treated with isavuconazole for invasive fungal infections. The study was undertaken despite the fact that the results of previous study on adults - SECURE have not indicated clear  clinical evidence for recommending routine therapeutic drug monitoring for isavuconazole. The main limitation of the present study is that the number of patients (26) is not sufficient to perform a meaningful population analysis. Thus, the results of analyses that may be interesting for clinicians should be taken with caution.  Moreover, some other issues need clarifications:

- Patient demographic data, diagnosis, and drug dosages should be presented in a table. In the present form (sections 3.1 and 3.2), these data are difficult to follow.

- Why the PK parameters from ref. 18 were not included in Table 1 and discussed?

- The number of patients in Table 1 should be shown for each case.

- As the previous model (ref. 18) was built using NONMEM, it would be better for parameter  comparisons to use the same software.

- The parameter values obtained in this study  using population analysis should be compared and discussed with those available in the literature.

- Besides mortality, the MIC values might have been used as a measure of drug efficacy. The Authors should comment on this.

- The conclusion that “a peak concentration would permit estimation of AUC” is not supported by the results.

- The last sentence is unclear (can be combined to obtain what?).

- Supplementary materials were not attached. The values of PK parameters and the observed vs. predicted plot should be presented in the main text.

Round 2

Reviewer 3 Report

The Authors have satisfactorily addressed most of may concerns. The only problem is with the names of PK parameters. They are different in the text and Table 1. Moreover, Ka should be rather the (first-order) absorption rate constant; KCP0 -  the (first-order) distribution rate constant from the central to peripheral compartment; Ke0 is the elimination rate costant from the central compartment. In the link model (Sheiner LB et al., 1979) ke0 denotes the rate constant describing distribution to the biophase, thus the name is very misleading (here it would be better to use k10 or just ke). KPC0 should be the (first-order) distribution rate constant from the peripheral to central compartment. It is not clear why "0" is included in these parameter names.